# Fused Deposition Modelling of Polymer Composite: A Progress

**DOI:** 10.3390/polym15010028

**Published:** 2022-12-21

**Authors:** J Mogan, W. S. W. Harun, K. Kadirgama, D. Ramasamy, F. M. Foudzi, A. B. Sulong, F. Tarlochan, F. Ahmad

**Affiliations:** 1Institute of Postgraduate Studies, Universiti Malaysia Pahang, Gambang, Kuantan 26300, Pahang, Malaysia; 2Department of Mechanical Engineering, College of Engineering, Universiti Malaysia Pahang, Gambang, Kuantan 26300, Pahang, Malaysia; 3Faculty of Mechanical and Automotive Engineering Technology, Universiti Malaysia Pahang, Gambang, Kuantan 26300, Pahang, Malaysia; 4Department of Mechanical and Manufacturing Engineering, Faculty of Engineering and Built Environment, Universiti Kebangsaan Malaysia, Bangi 43600, Selangor, Malaysia; 5Department of Mechanical and Industrial Engineering, College of Engineering, Qatar University, Doha P.O. Box 2713, Qatar; 6Department of Mechanical Engineering, Universiti Teknologi Petronas, Seri Iskandar 32610, Perak, Malaysia

**Keywords:** FDM, polymer, composite, properties, process parameter, application

## Abstract

Additive manufacturing (AM) highlights developing complex and efficient parts for various uses. Fused deposition modelling (FDM) is the most frequent fabrication procedure used to make polymer products. Although it is widely used, due to its low characteristics, such as weak mechanical properties and poor surface, the types of polymer material that may be produced are limited, affecting the structural applications of FDM. Therefore, the FDM process utilises the polymer composition to produce a better physical product. The review’s objective is to systematically document all critical information on FDMed-polymer composite processing, specifically for part fabrication. The review covers the published works on the FDMed-polymer composite from 2011 to 2021 based on our systematic literature review of more than 150 high-impact related research articles. The base and filler material used, and the process parameters including layer height, nozzle temperature, bed temperature, and screw type are also discussed in this review. FDM is utilised in various biomedical, automotive, and other manufacturing industries. This study is expected to be one of the essential pit-stops for future related works in the FDMed-polymeric composite study.

## 1. Introduction

Manufacturing industries are rapidly evolving in terms of the technology and materials involved. AM has transformed the industries of affordable three-dimensional (3D) solid structure fabrication and rapidly converting computer-generated designs into actual parts [1,2]. In recent years, AM has emerged as one of the most effective processes where the material is printed layer-upon-layer for building 3D products. Rapid prototyping, rapid manufacturing, and 3D printing are terms used to describe AM, which is snowballing in the manufacturing sector because the product can be served directly to the consumers, resulting in lower capital expenditure and transportation costs. Furthermore, AM fabricates customised parts in small quantities, which do not need special tools and allow the fabrication of complex geometries and assemblies [3,4]. In addition, 3D printing technology has advanced rapidly in recent years, and now various field applications are available, such as industries in the biomedical [5,6], aerospace [7,8], and automotive fields [9]. Unlike the traditional manufacturing processes [10], 3D printing is an AM method that works by stacking material from one layer over another to produce complex structures [11].

Metals, polymers, and composites can be used in AM. Many applications use AM technologies to create a complex shape [12]. There are many AM processes, such as selective laser sintering (SLS) [13], fused deposition modelling (FDM) [14], direct metal deposition (DMD) [15], laminated object manufacturing (LOM) [16], ink jet modelling (IJM) [17], and stereo-lithography (SLA) [18]. These procedures differ in printing material, process parameters, precision level, and end-use application [19]. FDM, also known as fused filament fabrication (FFF), is very well used in 3D printing. Stratasys Inc. in the United States invented the method during the 1990s. Printing factors such as printing orientation, air gap, layer thickness, raster width, and raster angle can be adjusted to improve the quality of printed parts [20]. Although FDM is known for its low operating cost and low investment cost, the printed products are more fragile as compared to other standard plastic manufacturing methods, such as moulding, injection [21], CNC [22], and extrusion [23,24].

Composites are materials of two or more physically or chemically separate phases divided by a discrete interface. The different elements are deliberately merged to produce a system with much more effective structural and functional properties than any of the constituents could achieve on their own [25].

Whether natural or synthetic, polymer composites are amongst the most significant applications of polymer. In various polymer matrices, the polymer composite is a multiphase solid substance in which one phase has at least 1, 2, or 3 dimensions. The polymer composites are viable for use as a high-performance composite when the reinforcing properties differ significantly from or exceed that matrix. The polymer matrix composite is the most enhanced composite material; these composite materials have various classifications of fibres such as natural and synthetic fibres as the reinforced materials in various types of polymer, for instance, thermoplastic polymer or thermoset polymer, which can be moulded into various shapes and sizes to produce various types of antiquated material [26].

By applying the feeding force created by the grooved bearing and driving gear, material in the form of a filament is fed into the liquefier head over the spool, as shown in Figure 1. The thermoplastic filament is heated to a glass transition temperature before being deposited in layers using a heated nozzle. The head of the liquefier travels through the X-Y plane according to the tool path supplied by programming. Support material can be eliminated with a solvent after fabrication [27].

Figure 2 shows the steps involved in the FDM process. The process begins with designing a digital model of a part by using CAD software. Then, 3D scanning, and reverse engineering are also performed to create a digital model. Later, the digital model is converted into a Standard Tessellation Language or Standard Triangle Language (STL) file. The STL file contains data about the surface geometry of the model. Then, the STL file is fed into the slicer software after conversion. Slicing determines the condition of printed pieces. Next, they apply information from the STL file, and the slicer generates G-codes. The generated G-code is the same as the CNC machine. It also controls the extruder and platform’s direction during printing. After converting the G-codes from an STL file, the 3D printer is ready to print a physical object of the design. This printing differs regardless of the kind of AM technique used. In the FDM process, the nozzle follows the G-code instructions and moves to deposit the melted filament in layers. The G-code controls the amount of material extruded, movement of the extruder nozzle, and extrusion time. After the whole model is printed, some post-processing is required to ensure a satisfactory product finish [28].

Even though the FDM process generates high-quality machinable materials, the world currently requires far more effective 3D printed parts than the traditional approaches, which can be gained by combining polymer with other materials, such as carbon, ceramic, metal, and many others, which are known as polymer composites. This review paper discusses the polymer composite using the FDM process from 2011 to 2021. As for this review paper, around 150 high-impact related research articles have been analysed to obtain the related information by using a research matrix table method. The paper also discusses polymer composite utilisation in the FDM process and its properties, as well as processing parameters and applications of the FDM process used in various industries. The dimensional accuracy of FDM-printed prototypes is one of the many aspects that determine the performance of fabricated prototypes because it influences the outcome of further prototype investigations. In addition, many printing parameters such as extrusion temperature, layer thickness, printing speed, raster width, and infill pattern are proven to significantly impact dimensional accuracy [29].

## 2. Composite Materials

Composite materials have developed into engineering materials that are remarkable and diverse because they are strong in domain areas that do not deform easily and have a high strength-to-weight ratio [30,31]. A composite material is defined by its name, which implies that it is made of different materials. Compositional engineering occurs when many constituent materials with significantly different chemical or/and physical properties are combined to form a new material with unique characteristics that are not present in the individual element [32,33,34,35,36,37]. Compared to the qualities of individual materials, this augmentation makes composite materials preferable. There are different types of composite material: scale base composite, reinforced base composite, matrix material base composite, and bio-composite. Figure 3 summarises the types of composite material. A composite material comprises two materials, namely base and filler. Because it wraps and bonds the reinforcement of other materials, the base material is commonly known as a matrix or a binder material. Fibres, particles, fragments, and natural or synthetic whiskers are filler materials [38,39,40]. Matrix is a soft phase with mechanical and physical properties, such as formability, ductility, and thermal conductivity. Material with high stiffness, strength, and low thermal fluctuation is included in the matrix reinforcement. The reinforcement phase of composites is always stiffer and more robust than the matrix because it conveys the load applied to the material.

### 2.1. Polymer Matrix Composite

A polymer matrix composite (PMC) is a composite material consisting of a natural polymer grid that holds a series of smaller uniform filaments. PMCs are designed to transport loads between matrix material filaments. PMCs are composed of a thermosetting or thermoplastic matrix with carbon, Kevlar, glass, and metal fibres dispersed throughout [41,42,43,44]. Thermosets are more commonly used than thermoplastics because of their increased strength and tolerance to high temperatures [45]. Thermosets are made by combining resins that are hardened together. The most common type of laminar structure is created by stacking and bonding thin layers of fibre and polymer until the appropriate thickness is achieved. Due to facile handling procedures and cheap manufacturing methods, PMCs are inexpensive composites [46,47]. Polymers can be utilised as the base matrix. Metals, generally in powders, are often used as the reinforcement, resulting in a material with unique qualities. The PMC has proliferated in recent years due to the demand for additional innovative engineering materials with higher strength and lower weight [48,49]. Figure 4 explains the classification of matrix composites. There are three types of matrix composites: metal matrix composites (MMC), ceramic matrix composites (CMC), and polymer matrix composites (PMC).

### 2.2. Base Material

AM and eventually the fabrication of original equipment manufacturer (OEM) components rely heavily on polymer materials. There are two kinds of thermoplastic materials in use today, which are thermoplastic and thermoplastic composites. The use of large-capacity polymer thermosets and elastomer materials in AM is a relatively new technique. FDM technology uses a variety of thermoplastics as feedstocks, including acrylonitrile butadiene styrene (ABS), polylactic acid (PLA), polycarbonate (PC), polyether ether ketone (PEEK), polyethene terephthalate glycol (PETG), and nylon [50,51]. Table 1 shows thermoplastics used in FDM process.

Figure 5 shows the base materials used in the FDM process. The most used base material is ABS, and the second most used is PLA. Both are thermoplastic materials that are commonly used materials in the FDM process. Comparatively, materials other than ABS and PLA are used at a minimum, at no more than 10%. PC, PEEK, and PETG are thermoplastics primarily utilised in engineering applications. ABS is a popular thermoplastic material for the FDM process because ABS has excellent melt fluidity, strength, and stiffness.

FDM ABS products’ impact and tensile properties are poor compared to injection-moulded components [52]. FDM-manufactured ABS products have a 34% lower tensile strength than injection-moulded products. PLA is another popular thermoplastic material utilised in the FDM process because of its biodegradable qualities and wide range of uses in the medical sector. PLA has low ductility as compared to ABS. However, it has high strength. During the printing of the PLA composite, a rise in void content and anisotropy is observed, similar to ABS composites in the FDM process [53].

**Table 1 polymers-15-00028-t001:** Thermoplastic used in FDM process.

References	Type of Materials	Characterisation	Utilisation Sector	Remarks
[54,55,56,57,58]	Acrylonitrile butadiene styrene (ABS)	Better resistance to corrosive materials Low cost Withstand high temperature	Microdevices Microfluidics Prototyping	Dissolves in acetone
[59,60,61,62,63,64,65]	Polylactic Acid (PLA)	Low cost Non-toxic Biodegradable Ease to print	Tissue engineering Automotive Electrical and Electromagnetic Biomedical Biosensors Prototyping	Very brittle Low toughness
[66,67,68]	Polyamide/Nylon	Resistant to impact Heat-resistant High tensile strength	Fabrication tools Prototyping Industrial production parts	Moisture accumulation
[69,70,71,72]	Polycarbonate (PC)	Transparent Temperature-resistant High resistance to impact	Dental Tissue engineering Orthopaedic	
[73,74]	Thermoplastic polyurethane (TPU)	Good lubricity Abrasion-resistant	Hoses and tubes Biomedical prototype Seals and gaskets	Elastomeric behaviour
[24,75,76,77,78]	Polyether-ether-ketone (PEEK)	Organic thermoplastic polymer Chemical-resistant Good lubricity	Aircraft parts Racing cars Drones Medical implants	
[79,80,81,82]	Polyethene terephthalate glycol (PETG)	Chemical-resistant Transparent High processability	Bone models Orthopaedics	Become brittle due to heat
[59,64]	Polyvinyl alcohol (PVA)	Soluble in water	Dental models Bioprinting Brackets	Affected by humidity
[83,84]	Polyetherimide (PEI)	Chemical-resistant Heat-resistant Dielectric	Aerospace Automotive Medical	Better than conventional plastic products

### 2.3. Filler Material

Plastics can be reinforced with a variety of fillers, including metals and numerous organic compounds from plants, to create composites. These fillers can be used to increase a composite’s characteristics, surface appeal, sustainability, or cost. Figure 6 shows filler materials used in the FDM process based on polymer composite-related published work. Carbon is the most used filler material (39%). The commonly used carbons are carbon nanotubes, graphite, graphene, and carbon black. There are a few carbon nanotubes: single-walled carbon nanotubes (SWNT) and multi-walled carbon nanotubes (MWCNTs). SWNTs are cylindrical graphitic tubules with diameters of approximately 1.0 nm. MWCNTs are a unique structure of carbon nanotubes, whereby the multiple single-walled carbon nanotubes are enclosed inside each other. Fibres can be classified into two categories: natural fibre and synthetic fibre. The natural fibre is extracted from animals, cellulose, and minerals. Fibre from minerals is asbestos. Animal fibres are silk, hair, and wool.

In contrast, cellulose is usually extracted from bast, leaf fruit, wood, seed, grass, and stalk. There are two divisions of synthetic fibre: organic fibre and inorganic fibre. Organic fibre consists of polyethene, aromatic polyester, and aramid fibre. Inorganic fibres are glass, boron, carbon, and silica carbide [26]. Although there is much information about continuous fibre-reinforced thermoplastic composites, there is not much on chopped carbon fibre-reinforced thermoplastic composites. This employs the FDM process to create composites and explore the impact of chopped carbon fibre on the thermo-mechanical properties of PLA composites [27]. Table 2 shows the filler materials in FDM process to enhance properties of polymer.

## 3. Processing Parameter of FDM

The process parameters influence the material’s accuracy, efficiency, and characteristics. As a result, a fundamental study into numerous process factors must be included to produce functionally efficient parts by utilising the FDM technology. Therefore, the FDM printing process specifies and briefly describes several parameters.

### 3.1. Layer Height

Figure 7 shows the bar chart for layer height used in the FDM process 3D printer. Nearly half of the 100% for layer height is 0.2 mm, consisting of 48%. Most FDM printers only have a size up to a 0.4 mm printing nozzle. The least-used layer height is 0.5 mm. The layer height is also defined as layer thickness. It means the thickness of material extruded from the printing nozzle for printing the physical part. The layer height can be adjusted to the printed parts’ referenced thickness. It represents the number of layers formed in a single pass all along the vertical axis of the FDM machine. Material deposition heights will be smaller than the nozzle diameter of the extruder.

The value is solely dependent on the diameter of the extruder tip. Layer height has an unavoidable impact on the impact and bending properties of the fabricated product. A minimum layer thickness is recommended to obtain better bending properties, and to increase layer thickness as it improves impact properties [139,140]. Compared to other parameters such as shell thickness and part orientation, the impact of layer thickness, as discussed in the literature, contributes roughly 85% of FDM-produced parts’ accuracy [141]. Based on the ANOVA results, other studies also show its significance (12.23% contribution) following the raster width parameter. The part dimension and layer thickness are determined to have a direct correlation. This indicates that thicker layers yield larger pieces, resulting in more significant dimensional variances [142].

### 3.2. Nozzle Temperature

Extrusion temperature is the temperature controlled inside the heating nozzle of FDM even before the material is extruded [143]. It changes the viscosity of printing material, affecting the part’s properties. The ideal temperature must be maintained since it can impact the viscosity of the filament material, which affects the printed part. Figure 8 shows the various nozzle temperatures used for the FDM process. The highest temperature used is from 230 °C to 259 °C. The second-highest range is 200 °C to 229 °C. Others stand for ranges from 380 °C to 409 °C, 410 °C to 439 °C, and 440 °C to 469 °C. As the material is extruded from the nozzle, the internal tension develops as the temperature of the material cools from the initial temperature to the temperature of the chamber. This occurs as a result of variations in deposition speed. The internal stress can cause interlayer and intralayer deformation, leading to manufactured part failure [144]. The nozzle temperature melts the filament into a semi-liquid state to print the physical parts. The extrusion temperature would be an essential parameter because if the temperature were low, the material would have a high viscosity which would be hard to extrude. However, if it is too high, the substance will flow, and dripping might occur. As a result, it is vital to fix the extrusion temperature to the correct value, depending on the material used for printing [141].

### 3.3. Bed Temperature

Besides nozzle temperature, bed temperature also plays a vital role in printing. Bed temperature, commonly known as heat bed, is a platform whereby the part is printed during printing. Bed temperature has two primary purposes. Firstly, it can prevent the printing object from warping. Warping is a familiar problem when the edges of the printed material are cooled at various rates compared to the rest of the material. When a heated and stretched material is extruded onto the cold and contracted material, it causes tensions in the material after the new layer cools. This causes the cooled plastic to warp upwards and changes the appearance of the print. Secondly, it helps in layer adhesion. It increases the surface energy of the print bed to improve the bonding strength of the first layer. The prints will not adhere properly to the build plate if the first layer adhesion is poor, which increases the possibility of a print failure. When it has good adhesion, it helps to reduce the warping of the printing material. Besides preventing warping and layer adhesion, the bed temperature also retains the temperature of the printing platform. The bed temperature also eases the removal process of printed parts. The removal process is straightforward with the bed temperature by using any cutting tools or forces. Figure 9 shows a column chart for the bed temperature of the FDM process. The highest range used is from 50 °C to 70 °C with 39%. The second primarily used range lies from 100 °C to 124 °C, with 18%. The minor range used is 125 °C to 149 °C. Others are ranged from 175 °C to 199 °C and from 200 °C to 224 °C. 

### 3.4. Printing Speed

In 3D printing technology, print speed is the most critical setting. As the name implies, print speed determines how fast the motors of the printer move. The electric motors controlling the X- and Y-axes and the extruder are included in the printing speed. The speed of the nozzle represents the deposition of filaments over a region of the built component during deposition. Printing speed is equal to the amount of time taken to print. It has a significant influence on the quality of the fabricated model. However, in narrower layer printing, the impact of the printing speed is negligible [145]. Figure 10 shows the printing speed used in the FDM process. The frequently used printing speed is around 21 mm/s to 40 mm/s. The next is 41 mm/s to 60 mm/s.

### 3.5. Building Orientation

Building orientation is also known as building direction. Besides the layer resolution, the build orientation is also very crucial to the process. The building orientation is the angle at which the part is placed about the horizontal axis of the build platform. Surface roughness and staircase effect are determined by resolution, whereas print quality and layer arrangement are determined by build orientation, and fusion is proportional to the mechanical properties of the printed object [14,146]. It describes how parts are positioned on the build platform about the three primary axes of the machine tool, which are the X-axis, Y-axis, and Z-axis [147]. Figure 11 shows the printing orientation of the impact test sample for 0°, 45°, and 90°. Afrose [148] observed that specimens’ best strain energy storage capacity and fatigue life are printed at a 45-degree angle.

### 3.6. Screw Type

#### 3.6.1. Single Screw Extruder

The single screw extruder was invented in the 1870s. It is the most extensively used extruder due to its ease of operation in polymer and rubber production [149]. The most basic single screw extruder configuration is a single revolving screw positioned inside a static cylindrical barrel split into three distinct zones: the compression zone, feed zone, and metering zone [150]. Different pressures can be generated along the length of the screw by varying the depth and pitch of the screw flight within every zone. The screw flight pitch and depth are generally chosen at bigger scales than from other zones to achieve low pressure at the feed zone, which consistently feeds material from the hopper into the extruder barrel [151]. Solid materials must be melted and homogenised as part of the rotating conveyance into the compression zone to form a suitable shape for distribution in the metering zone. As a result, the decreasing screw flight and pitch depth cause a progressive increase in barrel pressure along the length of the compression zone [152]. For less demanding applications, kneading, devolatilising, and mixing can be conducted in this processing zone [152,153]. A homogenous product by consistent structure can be achieved from die extrusion by stabilising the effervescent flow to a steady condition in the last metering zone. Fabricating extrusion-based goods using a single screw extruder is particularly well suited for high viscosity polymers since it may produce high pressure during operation [154].

The material is fed into the barrel through the feed throat by gravity force, and in most single screw extruders, the speed of the screw controls the output rate. High pressure to transfer material from the feed system causes the plastic pellet or powder to condense into a solid bed [155]. Periodically, the mass flow rate is unaffected by the speed of the screw and is regulated directly from the feed system by using a starving fed mechanism, resulting in an output rate that is lower than the forwarding efficiency of the screw [156,157]. The single screw extruder has only one screw and is used to make homogeneous polymers in a continuous shape [149,158]. Single screw extruders are unsuitable for heat-sensitive polymers due to higher friction and thermal energy as the screw speed increases.

Furthermore, significant pressure is used during the extrusion process, compressing the ingredients to generate filaments. However, it may lead to agglomeration and poor mixing due to a lack of shear deformation. [159,160,161].

#### 3.6.2. Twin Screw Extruder 

Although the single screw extruder process seems simple and inexpensive, it lacks the mixing capacity required to produce a polymer composite using many compounded components. As a result, in the late 1930s, a modified extruder known as a twin screw extruder that contained two screws placed next to each other at a modular barrel was developed to form intimate blends of two or more different materials [162]. Unlike a single screw extruder, a twin screw extruder provides a more vital shear force amongst the screws and barrel and the rotating screws, resulting in the proper mixing of materials [163]. As a result, a broad range of mixing operations and heat transfers are achieved with a faster throughput, independent of the screw’s speed. On the other hand, the counter-rotating system can generate a sizeable extensional shear force between the gaps between the two screws, allowing for a significant potential air entrapment, pressure generation, and extended retention period while using the minimal speed and output of the screw [164,165]. Both can be divided into two categories: entirely intermeshing and non-intermeshing. Because of its self-wiping function, the intermeshing twin screw extruder may not only eliminate non-motion throughout the extrusion, but also avoid excessive overheating of raw materials. After the process, the rotation of the screws removes residual material from the screw roots and cleans the entire inside barrel. Table 3 shows the difference of single screw extruder and twin screw extruder.

Meanwhile, this popular layout can help reduce product waste at the end of the manufacturing process [166]. The mutually different screws positioned in the extruder barrel for the non-intermeshing type result in low torque generation and weak interaction, making it a better choice for processing high viscosity materials and venting to eliminate interior volatile substances [167]. Figure 12 shows the type of screws.

**Figure 12 polymers-15-00028-f012:**
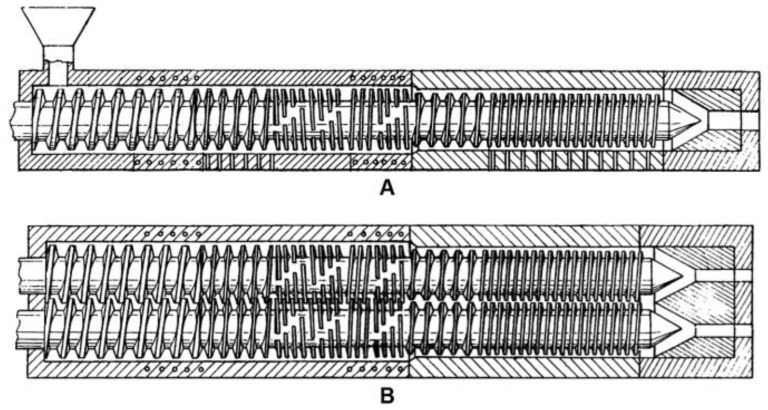
(**A**) single screw extruder and (**B**) twin screw extruder. Adapted with permission from Uitterhaegen. [168]. 2017, Elsevier.

**Table 3 polymers-15-00028-t003:** Single screw and twin screw are compared.

Extrusion Type (Screw)	Extruder Model	Advantages	Disadvantages	Reference
Single	SJ-30/25, Zhangjiagang Grand	Cheap Simple design Low maintaining cost	Poor in mixing Not suitable for low heat-resistant materials	[169,170,171,172]
Twin	SJ-30/25, Zhangjiagang Grand APV Chemical Machinery MP 2015 DSM Xplore Haake Rheomex OS, Thermo Fisher, Germany	High dispersion capacity, which results in better mixing Better process parameters control Easy material feed Flexible and better productivity	Expensive Better input energy Not applicable for materials that are shear-sensitive	[171,172,173,174,175]

## 4. Properties of FDM-Polymer Composite

### 4.1. Mechanical Analysis

Mechanical properties are testing aids in evaluating and designing materials and products, allowing them to last longer and be more efficient and cheaper. AM polymers’ characteristics are tested using ASTM and ISO test methods. They also aid in the creation of desired items. In order to prepare the sample and conduct mechanical experiments, research organisations use ASTM standard criteria; for example, practically all research groups evaluated for tensile tests [176,177] employ ASTM D638. The majority of research findings state that the component’s ultimate tensile strength, yield strength, elasticity, and elongation are mostly affected by the process parameters. Figure 13 shows the mechanical properties that were tested from 2011 to 2021. From the study conducted, the most tested mechanical property is tensile. Flexural is the second most thoroughly tested mechanical property. Fatigue behaviour is one of the mechanical properties that has undergone minor testing. The tensile test is used on samples to determine material parameters, such as ultimate tensile strength, yield stress, Young’s modulus, ductility, and toughness. There are many shapes and designs to choose from when producing a sample for the tensile test. Even though there are various options for the test sample design, it must adhere to ASTM standards [178].

The advancement of the fibre composite by using the FDM technique has expanded opportunities for field research. The tensile properties of composites are heavily influenced by factors such as printing parameters, fibre content, and fibre reinforcement. The percentage of fibre reinforcement and the orientation of fibres determine the tensile strength of the composite produced by the FDM process [179,180]. The interlaminar shear strength of the FDM-fabricated fibre composite is compromised, which directly impacts the flexural strength of composites. In addition, the FDM-fabricated fibre composite has anisotropic properties that cause variations in strain rate when bending, which increase the shear stress amongst the layers, and ultimately cause the separation of layers and failure. Therefore, the interlaminar shear strength is an essential factor to consider when improving the flexural properties of the FDM-printed fibre composites [181]. In order to significantly increase the mechanical qualities of FDM 3D-printed PLA items, such as average tensile strength and impact toughness, Kuan et al. [182] and Li et al. [183] created FDM printing filaments containing carbon fibre and MWCNT for the reinforced phase-modified PLA.

### 4.2. Thermal Analysis

Figure 14 shows a chart of thermal analyses. There are three types of thermal analyses: dynamic mechanical analysis (DMA), thermogravimetric analysis (TGA), and differential scanning calorimetry (DSC). The highest thermal analysis used is DSC with 41%, while TGA is 31%. Essential qualities of energetic materials are their thermal properties, which are strongly related to safety during manufacturing, storage, transportation, and usage. The thermal properties of energetic materials can be determined quickly, efficiently, and effectively using thermal analysis tools [184]. However, in thermal analysis investigations, the fluctuation of thermogravimetry (TGA) curves frequently occurs for unknown reasons, and the mass–loss curves, for example, grow or drop drastically and even surpass the complete scales of instruments. Furthermore, the differential scanning calorimetry/differential thermal analysis (DSC/DTA) is inconsistent with TG because the crucible frequently shifts or slips off the sample pan [185]. Table 4 shows the difference between TGA and DSC.

## 5. Application of Polymer Composite in 3D Printing

Various industries are using the FDM process to produce products. Figure 15 shows the FDM application in industries. The biomedical field is one industry that utilises the FDM process for its product, whereby the industry utilises 38% out of 100%. Manufacturing is the second-highest industry in utilising the FDM process, and aerospace is the third-highest industry. In recent times, the use of FDM technology has grown in popularity, particularly in aerospace, medical, and automobile fields. In addition, the overall quality of prototypes printed for the aerospace sector is in high demand since they are utilised to examine the fluid dynamic behaviour of models [141].

### 5.1. Aerospace

In the past few years, the production of aircraft, satellites, and space shuttles has drastically increased the demand for aerospace and aviation components [199]. Aerospace components with a high aesthetic value rather than high efficiencies, such as light housings, door handles, power wheels, and complete dashboard designs, are usually made using 3D printing. Metal 3D printing enables the manufacture and deployment of complex military components more quickly [200]. However, recycling scrap formation after manufacturing aircraft parts is costly and time-consuming. As much as 80–90% of the conventional billet may be wasted during machining, but an AM process can reduce this by less than 10%. AM also allows the creation of free-form designs that produce tooling fixtures for making expensive aerospace materials such as titanium. As a result, conventional manufacturing methods can only produce cooling channels with straight lines, complicating aerospace components’ fluid flow optimisation [201]. Figure 16 shows Stratasys and Aurora Flight Sciences’ AM-unmanned aerial vehicle (UAV). 

### 5.2. Automotive

An AM technique widely used is fused deposition modelling (FDM). It is used in the automobile industry for various purposes, including lightweight equipment, final functional components, and testing models. On the other hand, the FDM technology faces two significant challenges in becoming a viable processing method in the automotive industry, which are weak and anisotropic mechanical characteristics and a limited range of printing materials. The mechanical properties of FDM’s physical parts are influenced by weak interlayer links produced during the layer top layering process [203]. By design, the brake pedal is one of the most critical pieces of the vehicle. A brake pedal is a safety component with sound engineering, exact quality requirements, and critical quality inspection. In addition, the brake pedal is the most acceptable option for research because of its severe characteristics. As a result, if a crucial part can be produced with the AM technique, then less critical parts such as brackets, hinges, and supports, to name a few, can also be produced [9]. Figure 17 shows the brake pedal produced by using the FDM process.

### 5.3. Biomedical

Biomedical implants benefit the medical profession and end-users, such as people who have suffered severe accidents or illnesses. However, biomedical implants are artificial substitutes expected to function similarly to the original. Therefore, any substance used as an implant must be compatible with the human body [204]. Because of its ability to provide personalised fabrication at a minimal cost, 3D technology has piqued the interest of industrial and academic fields. In addition, 3D printing technology enables the creation of polymeric materials for biomedical applications due to inherent advantages, such as the capacity to create complicated geometry quickly. There are four broad study categories in which the latest 3D printing technology developed for medical application can be classified as the formation of diseased organs, development of permanent non-bioactive implants, and development of biodegradable and bioactive scaffolds on organ and tissue printing [205]. In addition, the FDM-fabricated bone prototypes can be used in biomedical research and real-world testing; these can be 3D solid models, specimens for similar mechanical prototypes, and mechanical testing. Geometric parameters, as well as local or global mechanical properties, can be evaluated by the models showed in Figure 18 [206].

### 5.4. Textile

Textile companies are also beginning 3D printing to fabricate dresses, shoes, and other items [207]. Good adhesion and stability are the primary factors in textile manufacturing [208]. Polylactic acid (PLA), acrylonitrile butadiene styrene (ABS), polyamide (PA), and polycarbonate (PC) are usually used in FDM. In addition, fibres, fillers, dyes, and other additives are commonly used to produce filaments [209,210]. The widespread application of PLA and Soft PLA is used to print smooth, glossy, soft, and lacelike fabric structures, because they are more flexible than PA and ABS and give the end product a soft handle [207]. On the flip side, ABS is rigid, making it suitable for joints [210,211]. According to Samit et al. researchers have constructed various knitted structures and weaves using the FDM process, such as woven fabric structures with weft knitted structures and visible stitches. They have also experienced using FDM printing to create garment panels, lace structures, and composite structures [207,209,211,212]. Figure 19 shows an FDM-printed structure for textile.

### 5.5. Functional Materials

Functional materials are typically defined as those that have specific inherent qualities and functions of their own. Topology optimisation is a popular and effective method for determining structural configurations for various performance types. Additionally, it is viable to incorporate it into additive manufacturing. Chen et al. have built topological structures with an effective zero Poisson’s ratio: a framework based on periodic unit cells with plus-minus Poisson’s ratios established. The topologically designed structures are then printed using 3D printing technology and short carbon fibre reinforced polyamide (SCF/PA) [213]. The auxetic effect, also known as the negative Poisson ratio (NPR), can be induced in a hexagonal honeycomb by changing the cell angles. To achieve better mechanical properties and functionality, continuous carbon fibre (CCF)-reinforced composites are designed and 3D-printed by Chen et al. [214]. Furthermore, for the planar lattice designs, a center cross-lattice with four outer-strip components is created using carbon fibre-reinforced polyamide composites via FDM by Chen et al. [215]. Auxetic geometries are then turned into high-performance composites via the 3D printing technique, reinforced with chopped carbon fibre (CF). The effects of polynomial ordering and CF incorporation on the mechanical properties are carefully investigated by Hu et al. [216]. Chen et al. have studied the compressive behaviours of 3D-printed CF-reinforced polyamide composite metamaterials with NPR [217].

## 6. Future Trend/Challenges of FDM-Polymer Composite

Figure 20 shows the trend of polymer composite utilisation in the FDM process from 2011 to 2021. The graph shows that the trend is increasing yearly. From the study conducted, the utilisation of polymer composite had a drastic hike in 2017 and 2021 compared to the other years.

The use of AM and other advanced manufacturing technologies such as the FDM process is ushering in a new era in manufacturing, whereby value chains are short, tiny, more localised and personalised, cooperative, and sustainable. Compared to the typical subtractive techniques, these are potentials that enhance resource efficiency, reduce waste of expensive metals such as titanium, and enhance the design for assembly methods to improve characteristics and lower costs. 

There are many signs to suggest that the FDM process will continue to be integrated into current industrial processes and human life as the technology develops and becomes more affordable. As a result, the FDM sector, comprising technology and material advancements, as well as related services, has grown at an exponential rate.

## 7. Conclusions

AM is one of the most significant accomplishments of the fourth industrial revolution. The uses of AM have significant growth in many industries. FDM is the most popular AM due to its endless benefits. FDM is a process that can produce various complex designs into physical objects at a lower cost as compared to traditional manufacturing. This article discusses:An overview of FDM process flow and polymer composite material properties, and the type of base and filler material commonly used in FDM.The printing parameters such as nozzle temperature, bed temperature, printing speed, building orientation, layer height, and screw type also play a significant role in the performance of the FDM-printed product. However, the relationship between printing quality and mechanical behaviour for the various types of materials used in FDM cannot be explained by the available data. There are currently no absolute laws and regulations that can be applied to help users improve the printing process to achieve the best printing results, because the same printing procedure can result in various printing outcomes if the material is different.Different polymers have different behaviours in terms of mechanical properties. Adding fillers to thermoplastic polymers can enhance the properties and strength of that particular polymer. Meanwhile, it is still rare to come across the improvement of process parameters for thermal, chemical, and dynamic mechanical properties.The application of the FDM process in industries such as the aerospace, automobile, textile, and biomedical sectors is also explained briefly. However, in terms of large-scale applications, FDM still cannot be used as a substitute for the traditional technique, such as injection moulding, when analyzing the mechanical work performance achieved. The main issues with the FDM printing are porosity, gaps between layers, and rasters produced in the FDM process.

## Figures and Tables

**Figure 1 polymers-15-00028-f001:**
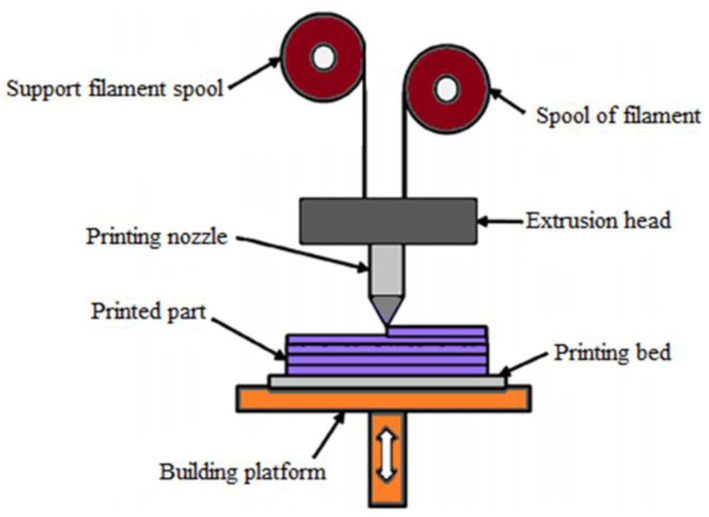
Schematic diagram of FDM process.

**Figure 2 polymers-15-00028-f002:**
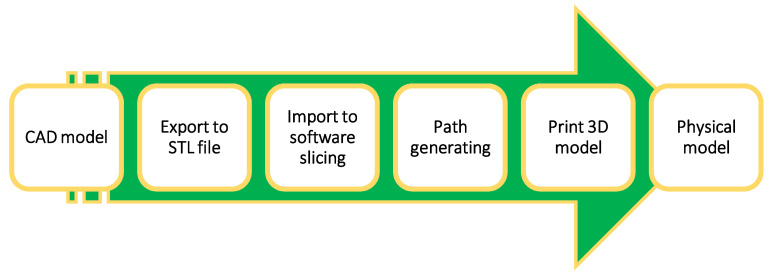
Steps involved in the FDM process to produce 3D printed parts.

**Figure 3 polymers-15-00028-f003:**
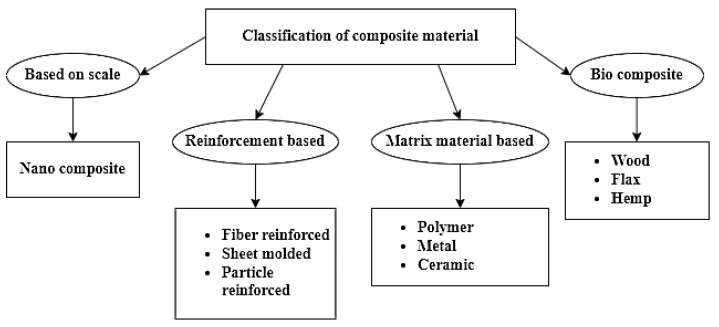
Detailed classification of composite materials.

**Figure 4 polymers-15-00028-f004:**
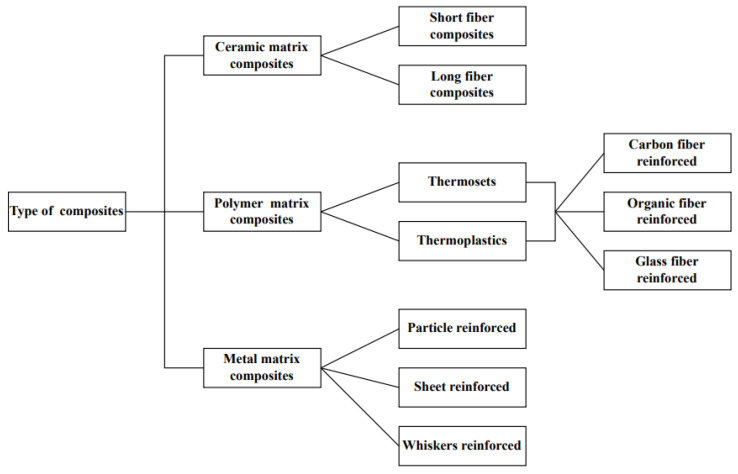
Classification of the type of matrix composites.

**Figure 5 polymers-15-00028-f005:**
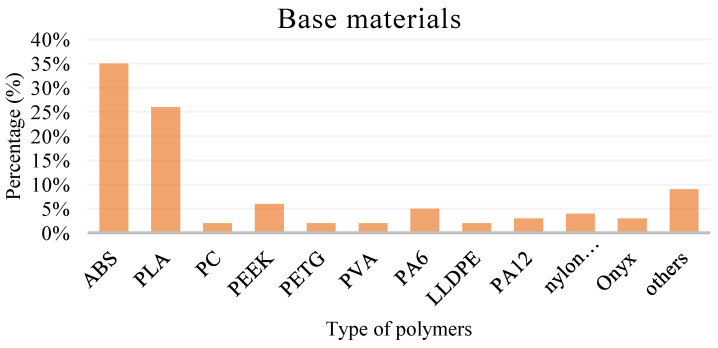
Type of polymer used as a base material in the FDM process.

**Figure 6 polymers-15-00028-f006:**
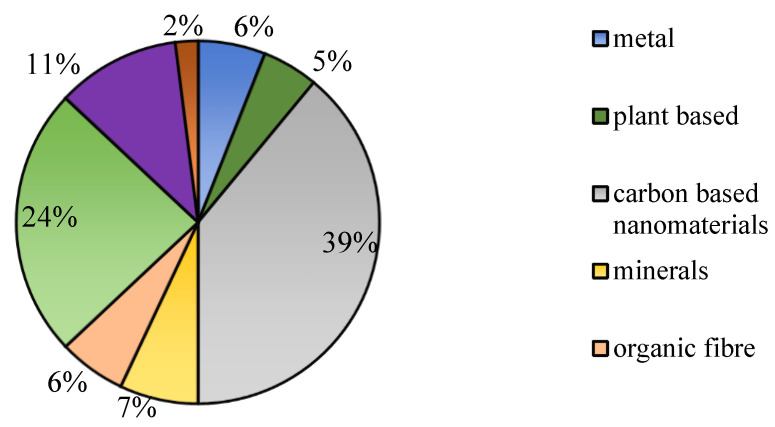
Materials that are used as filler to enhance properties of polymer.

**Figure 7 polymers-15-00028-f007:**
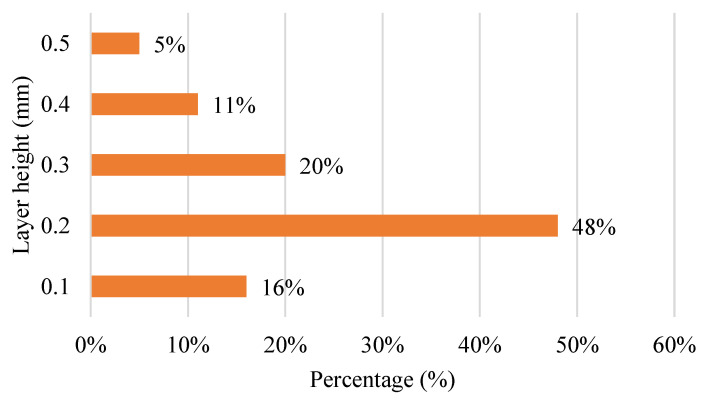
Layer heights used in FDM machine during printing process.

**Figure 8 polymers-15-00028-f008:**
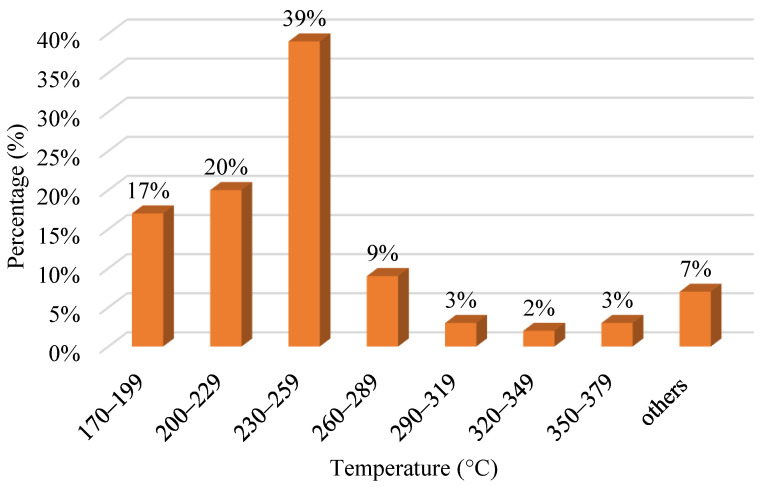
Range of nozzle temperature for various material via FDM process.

**Figure 9 polymers-15-00028-f009:**
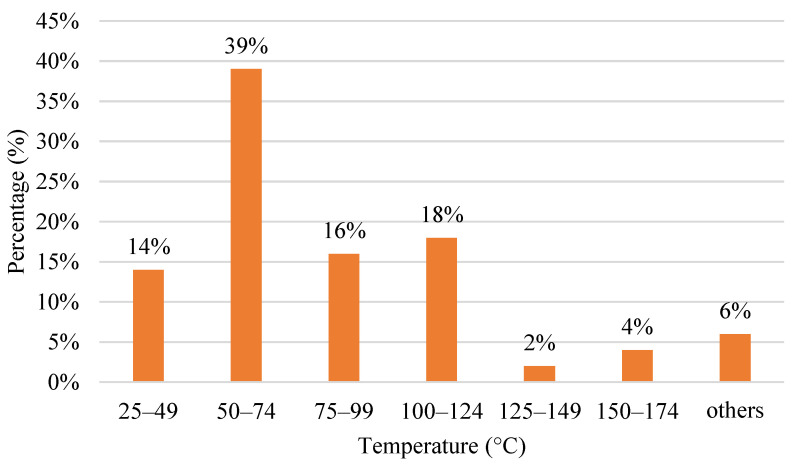
The range for bed temperature used in the FDM process for various material.

**Figure 10 polymers-15-00028-f010:**
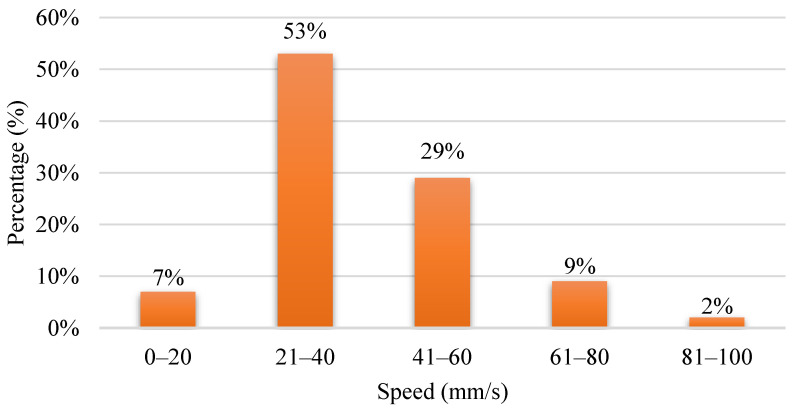
Range of printing speed used in FDM process.

**Figure 11 polymers-15-00028-f011:**
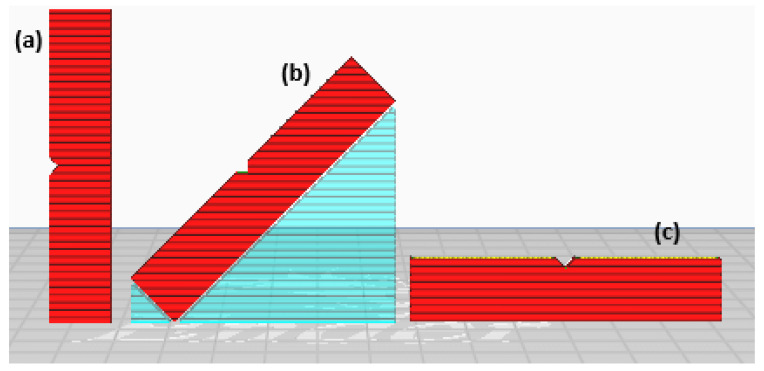
Impact samples printed with various (**a**) 90°, (**b**) 45°, and (**c**) 0° building orientations.

**Figure 13 polymers-15-00028-f013:**
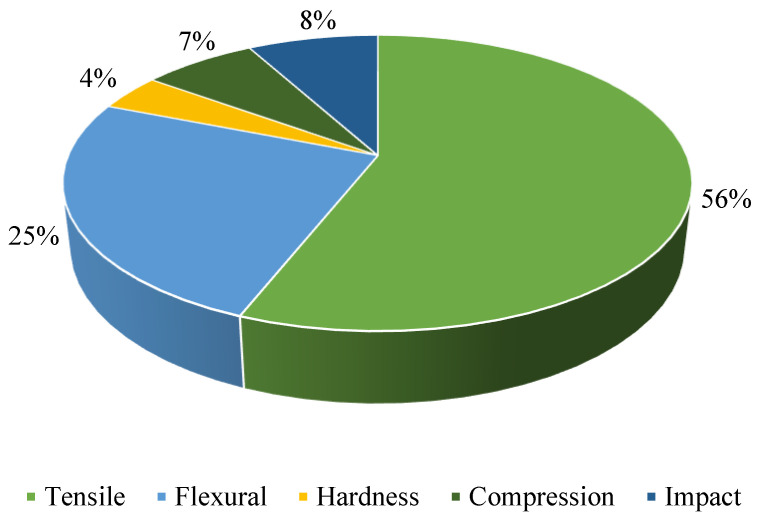
Mechanical properties that were studied primarily in the FDM process from the year 2011 to 2021.

**Figure 14 polymers-15-00028-f014:**
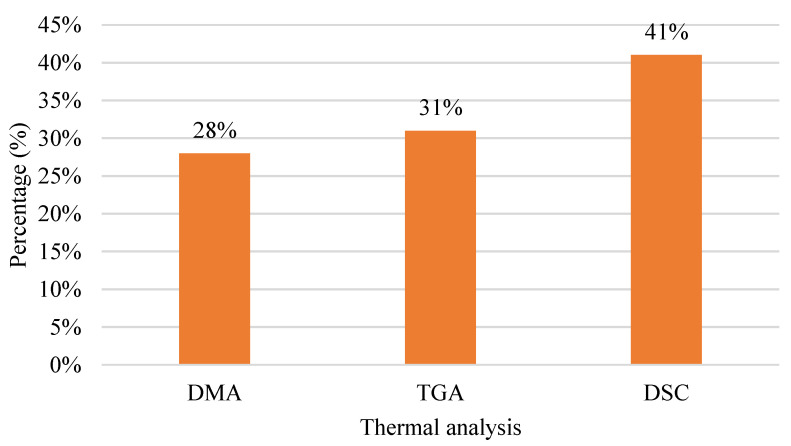
Thermal analysis of FDM polymer composite from the year 2011 to 2021.

**Figure 15 polymers-15-00028-f015:**
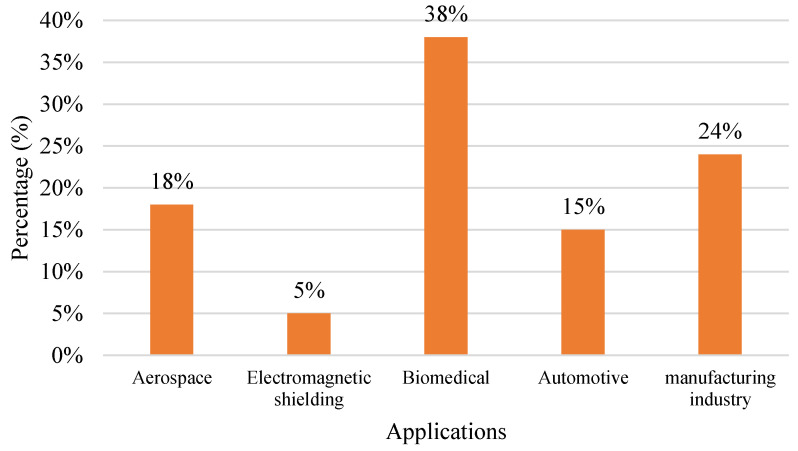
Application of FDM using polymer composite.

**Figure 16 polymers-15-00028-f016:**
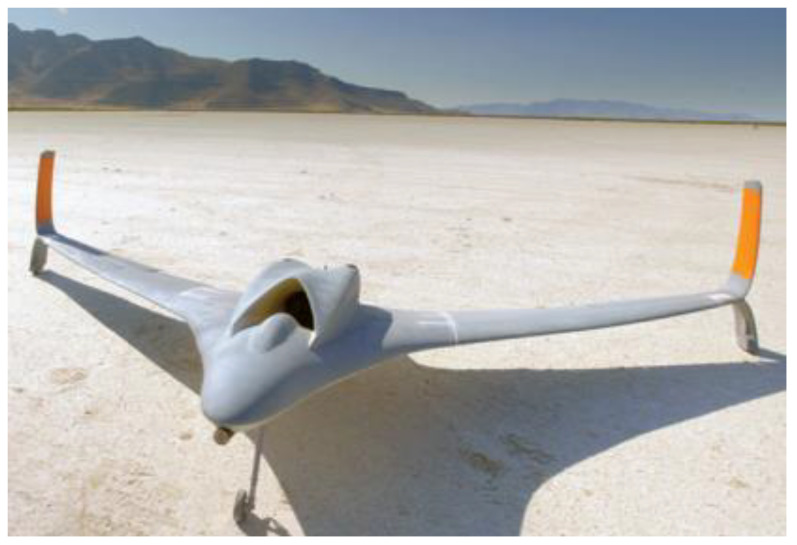
UAV using the FDM process. Adapted with permission from Stratasys [202].

**Figure 17 polymers-15-00028-f017:**
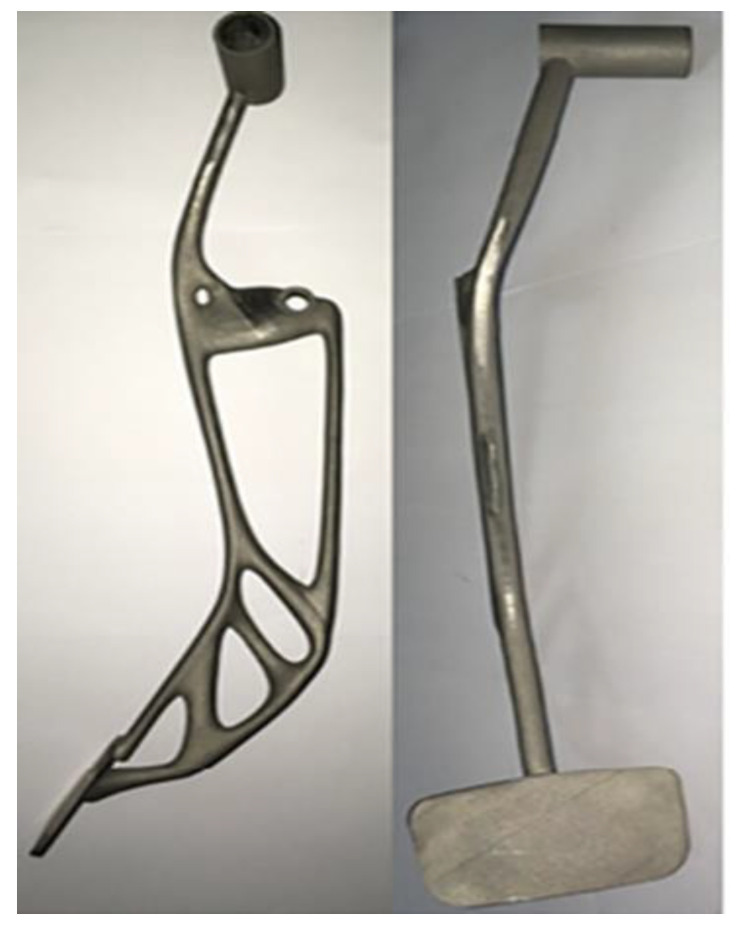
Metal brake pedal using BASF Ultra fuse 316L metal-polymer filament via FDM. Adapted with permission from Sargini et al. [9]. 2021, Elsevier.

**Figure 18 polymers-15-00028-f018:**
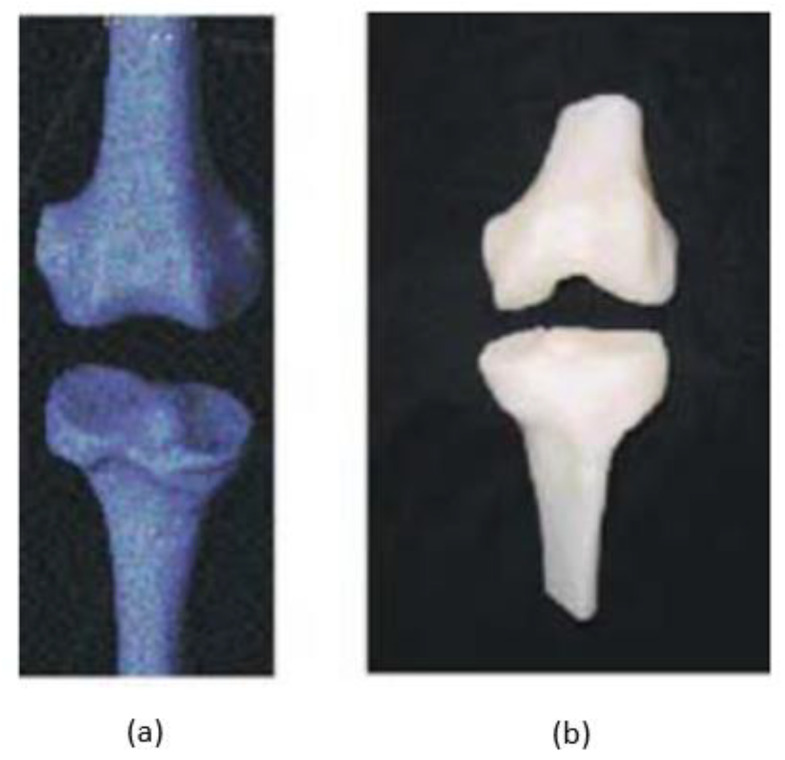
(**a**) Geometric model and (**b**) FDM prototype of femur and tibia. Adapted with permission from Revilla-León et al. [206]. 2002, IOP Conference Science.

**Figure 19 polymers-15-00028-f019:**
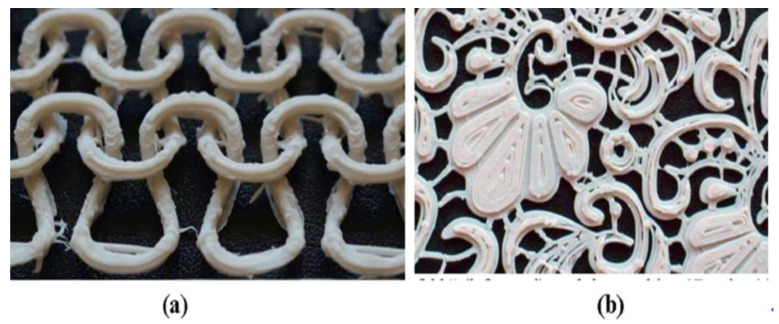
(**a**) 3D-printed weft knit and (**b**) lace structure in textile industry. Adapted with permission from Chakraborty et al. [212]. 2020, Elsevier.

**Figure 20 polymers-15-00028-f020:**
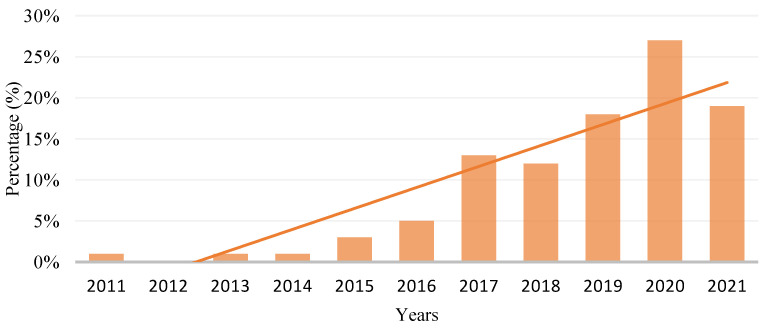
The trend of polymer composition utilisation in FDM from year 2011 to 2021.

**Table 2 polymers-15-00028-t002:** Classification of filler material and the conducted test.

Classification	Filler Material	Type of Base Material Used	Composition (wt%)	Test	Reference
Metal	Aluminium	PLA	6.95	Tensile	[85,86,87]
	Copper	PLA	4, 8, 12, 16, 20	Compression and flexural	[88,89,90]
	Stainless steel	-	-	Density measurement	[91]
Plant-based	Cork	PLA	5, 10, 15, 20, 25, 30, 50	Tensile and density measurement	[92,93]
	Wood particle	PLA	30, 40	Tensile, flexural	[94,95,96]
	Cellulose	PLA	1, 2, 5, 10, 20	Tensile, flexural	[97,98,99,100]
Carbon-based nanomaterials	Carbon black	ABS	3, 1.5	Density measurement, tensile	[101,102]
		PLA	5, 53	-	[103,104,105]
	Graphene	ABS	2, 4, 6, 8	Tensile, flexural, impact, hardness	[106,107,108,109]
		PC/ABS	0.2, 0.4, 0.6, 0.8	Tensile	[110,111]
		ABS/EPDM	2, 4, 6, 8, 10	-	[112]
	Carbon nanotubes	ABS	1, 3, 5, 7, 10	Tensile, density measurement	[101,113,114]
		PLA	10	Electrical conductivity	[115,116]
Mineral	Hydroxyapatite (HA)	PLA	5, 10, 15	Compression and flexural	[117,118]
		PCL	10, 20	Compression and tensile	[119,120,121]
		PEEK	10, 20, 30, 40	Tensile	[122]
Organic fibre	Kenaf bast fibre	PCL	5, 10, 20	Tensile and flexural	[123]
	Aramid fibre	Nylon polymer	2	Surface roughness	[124]
	Flax fibre	PLA	-	Tensile	[125]
Inorganic fibre	Carbon fibre	ABS	1, 2, 3, 5, 7.5, 10, 15	Tensile, flexural, surface roughness, dimensional accuracy	[126,127,128]
		PLA	12, 15, 20	Tensile, compression, flexural, hardness, impact	[129,130,131,132]
		PEEK	10, 20	Tensile, flexural	[133,134,135]
	Glass fibre	ABS	30 (vol%)	Tensile	[136]
		nylon	13.87 (vol%)	Tensile, flexural, impact	[137,138]

**Table 4 polymers-15-00028-t004:** Comparison of TGA and DSC.

	TGA	DSC	Reference
Primary determination	Changes in sample mass as a function of temperature or time	Changes in heat flow to and from a sample as a function of temperature or time	[92,186,187]
Temperature range	Room temperature to 1000 °C	−170 °C to 600 °C	[188,189,190]
Sample amount	Approximately 5–50 mg	Approximately 5–50 mg	[191,192,193]
Typical output	Lost or gained % by mass Residual mass	Transition temperature Transition enthalpy	[194,195]
Example of applications	Moisture content Decomposition Thermal stability Compositional analysis Oxidation	Phase transitions: melting and crystallisation Glass transition Solid–solid transitions	[196,197,198]

## Data Availability

Not applicable.

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
