# Peer review of "Fused Deposition Modelling of Polymer Composite: A Progress"

_polymers, 2022, doi:10.3390/polym15010028_

Round 1
Reviewer 1 Report
Harun et al. made a comprehensive summary on the fused deposition modelling of polymer composite in this review. This is clearly a hot subject of research worthy of a timely review. Generally, this review was well prepared with all important works on the reviewed field included. The useful information summarized in this work merits the publication in Polymers. The following minor concerns should be further addressed prior to acceptance.
1. The uniqueness of this review relative to the published reviews with a similar research topic remain unclear, which should be highlighted clearly in the abstract and introduction sections.
2. In the last section of this manuscript (Conclusion and perspective), please add more contents about the current problems that restrict commercialization of FDM, especially FDM-based various materials. In addition, please also point out future direction and solution to solve these problems. This will make this review more valuable and can provide the audience more information about what could be done in the future.
3. More details of results should be provided and highlighted in all figure captions for easy following and understanding for the readers.
4. Some grammar mistakes need to be checked and corrected throughout the manuscript.
Reviewer 2 Report
The article aims to review the state-of-the-art of fused deposition modelling for polymer composites. However, several big issues should be well addressed before publication. The reviewer’s comments are as follows:
(1) What is the biggest contribution of this review? Because several similar publications can be found currently. The reviewer cannot see the significance of this study which must be well presented.
(2) In abstract, “In the current scenario, additive manufacturing (AM) highlights developing complex and efficient parts for various uses. Fused filament fabrication (FFF) is the most frequent fabrication procedure used to make polymer products. FFF, also known as fused deposition modelling (FDM), is the most used AM method” must be well rephrased or deleted because it is well-known points. The abstract should address the key points of this study.
(3) “The base and filler material used and the process parameters include layer height, nozzle temperature, bed temperature, and screw type” is confusing, and this sentence must be revised.
(4) How to define bio composite in Figure 3? Will there be overlap between bio composite and other composites? Please clarify this point in the article.
(5) No introduction for Figure 6. Where does it come from?
(6) Applications are narrow which must be expanded to strengthen the importance and potential of FDM in composites. For instance, applications of polymer composites by FDM for functional materials or structures should be addressed. Here, some reference articles can be well discussed to emphasise this point (Thermal functional composites: Compos Sci Technol 2022; 227: 109599; Auxetic composites: Compos Part A 2021; 150: 106625; Int J Mech Sci 2021; 206: 106634; Quan C, et al. Compos Part B Eng 2020; 187: 107858; Short fibre reinforced functional composites: Hu C, et al. Compos Part B Eng 2020; 201: 108400; Compos Struct 2022; 293: 115717)
Reviewer 3 Report
The aim of the paper is to review the state-of-the-art literature about the use of polymer composites for FDM additive manufacturing applications. The study of composite polymeric materials is an interesting topic from a scientific and industrial point of view; however, unfortunately, the research developed by the authors presents certain limitations.
1.- The paper presents a review of the state-of-the-art literature related to composite polymeric materials for manufacturing components using the FDM additive technology. The paper focuses on presenting the current state of the art by analyzing the influence of manufacturing parameters of the FDM process and its influence on the part surface finish, mechanical performance, features of composites, and applications in the manufacturing FDM process. However, unfortunately, the authors don't detail the scientific methodology they followed to do the presented review. It is recommended that the authors detail the methodology employed to carry out the literature survey, indicating, for example, keywords, number of journals that have been analyzed, number of indexed papers that have been analyzed, number of Scientific Databases, etc.
2.- The authors present too general state-of-the-art conclusions without indicating future lines of work that help researchers to focus on unresolved problems at the research level.
Round 2
Reviewer 2 Report
The reviewer's comments have been addressed
Reviewer 3 Report
The authors have addressed satisfactorily the points raised during the review